

# Genetic and morphological diversity in populations of *Annona senegalensis* Pers. occurring in Western (Benin) and Southern (Mozambique) Africa

Janine Conforte Fifonssi Donhouedé[1,2,3], Isabel Marques[4], Kolawolé Valère Salako[3], Achille Ephrem Assogbadjo[1,3], Natasha Ribeiro[2] and Ana IF Ribeiro-Barros[4]

[1] Laboratoire d'Écologie Appliquée, Faculty of Agronomic Sciences, University of Abomey-Calavi, Cotonou, Benin

[2] Department of Forest Engineering, Faculty of Agronomy and Forest Engineering, Eduardo Mondlane University, Maputo, Mozambique

[3] Laboratoire de Biomathématiques et d'Estimations Forestières, Faculty of Agronomic Sciences, University of Abomey-Calavi, Cotonou, Bénin

[4] Forest Research Center (CEF), School of Agriculture, University of Lisbon, Lisbon, Portugal

Corresponding authors
Janine Conforte Fifonssi Donhouedé,
jdonhouede@gmail.com
Ana IF Ribeiro-Barros,
anaifribeiro@edu.ulisboa.pt,
aribeiro@isa.ulisboa.pt

## ABSTRACT

**Background**. Understanding morpho-genetic diversity and differentiation of species with relatively large distributions is crucial for the conservation and sustainable management of their genetic resources. The present study focused on *Annona senegalensis* Pers., an important multipurpose wild plant, distributed exclusively in natural ecosystems but facing several threats. The study assessed the genetic and morphological diversity, structure, and differentiation of the species in populations from Western (Benin) and Southern (Mozambique) Africa. The material was evaluated to ascertain the environmental (climatic) determinants of the variation within this species.

**Methods**. Four sub-populations comprised of 154 individuals were phenotyped based on nineteen plant, fruit, and leaf morphological traits and further genotyped using ten polymorphic nuclear microsatellite (nSSR) markers.

**Results**. The results indicated strong differences in plant, fruit, and leaf morphological traits between Western and Southern populations. Furthermore, the studied populations were characterized by high genetic diversity, with an average genetic diversity index of 1.02. Western populations showed higher heterozygosity values (0.61–0.71) than Southern populations (0.41–0.49). Western and Southern populations were clearly differentiated into two different genetic groups, with further genetic subdivisions reflecting four sub-populations. Genetic variation between regions (populations) was higher (69.1%) than among (21.3%) and within (9.6%) sub-populations. Four distinct morphological clusters were obtained, which were strongly associated with the four genetic groups representing each sub-population. Climate, mainly precipitation and temperature indexes, explained the relatively higher variation found in morphological traits from Western (40.47%) in relation to Southern (27.98%) populations. Our study suggests that both environmental and genetic dynamics play an important role in the development of morphological variation in *A. senegalensis.*

## INTRODUCTION

The change in land use and climate are fragmenting the natural habitats of many useful wild edible fruit trees (*Anuragi et al., 2016*). As a consequence, some of these species are threatened and have a narrow or fragmented distribution (*Chichorro, Juslén & Cardoso, 2019*; *IUCN, 2022*). Sustainable management and conservation of such useful species require a better understanding of the existing diversity to utilize their potential efficiently. However, such information is available for only a limited number of species, and many species are yet to be documented.

Population diversity quantifies the magnitude of genetic and morphological variability within a population (*Hughes et al., 2008*). The more diverse a population is, the more it can adapt to a changing environment (*Sheidai et al., 2014*). Morphological traits have been used as a tool to characterize the unexplored potential of germplasm for developing elite genotypes, *i.e.*, more resilient, productive, and nutritive (*Folorunso & Modupe, 2007*). Yet, the morphological variability observed in wild populations is usually the expression of the signal of genetic diversity shaped by environmental conditions. For instance, the morphological variability of *Prunus serotina* Ehrh was influenced by temperature and precipitation extremes (*Guzmán, Segura & Fresnedo-Ramírez, 2018*). Likewise, *Vitex doniana* Sweet was influenced by environmental traits, mainly climate factors (*Hounkpèvi et al., 2016*). However, the morphological variability found in *Polygonum aviculare* L. s was reported to rather have a strong genetic basis (*Mosaferi et al., 2015*). Therefore, although both genetic diversity and environmental conditions can drive variation in the observed phenotypes, their relative importance varies across species.

*Annona senegalensis*, also known as the wild custard apple, is an edible fruit plant widely distributed in Africa (*Orwa et al., 2009*). Its distribution spans South Africa, Mozambique, and Botswana (Southern Africa), and Benin, Niger, Burkina-Faso, and Mali (Western Africa). *Annona senegalensis* is a perennial woody, anemophilous, and predominantly outcrossing plant (*Kwapata et al., 2007*). It is a diploid species from the Annonaceae family, one of the largest tropical and subtropical families. It has a high nutritional, medicinal, and economic importance for African rural communities, contributing significantly to household livelihoods and income (*Mapongmetsem, Kapchie & Tefempa, 2012*; *Donhouedé et al., 2022*). Different parts of this species are also used in traditional medicine to treat diseases such as tuberculosis, gastritis, and snake bites, among others (*Okhale et al., 2016*). As a traditional food plant in Africa, *A. senegalensis* plays an important role in the context of food security, and its domestication has the potential to improve nutrition, foster development, and support sustainable land use. However, *A. senegalensis* is facing several threats due to its high exploitation, as well as land use changes that have resulted in severe degradation of its habitat (*Kwapata et al., 2007*; *Ba, Diémé & Sy, 2021*). Despite several past studies have highlighted that this species will likely disappear without any

conservation efforts (*Campbell & Popenoe, 1988*; *Kwapata et al., 2007*; *Ba, Diémé & Sy, 2021*), genetic data that would assist in this procedure are still largely missing. Only one study assessed the genetic diversity of *A. senegalensis* and it was based on only three microsatellite markers and three populations occurring in Malawi (*Kwapata et al., 2007*). In Western Africa, some authors reported high morphological variability in *A. senegalensis* populations and attributed 42% of this variability to climate (*Hounkpèvi et al., 2020*). Whether such morphological variation can still occur in a larger geographical range is unclear. Furthermore, understanding if the observed role of climate at the local scale can be expanded to a larger geographical range is essential to better follow the species response to environmental conditions. Species with a wide range of distribution often grow under diverse environmental conditions which give an opportunity to study how genes are expressed and the probable response of their populations to future climate change. The use of molecular markers is known as one of the best tools to study genetic material and explore genetic diversity in plants (*Feng et al., 2016*). Simple sequence repeats (SSR) or microsatellite markers are codominant, easily automated, highly polymorphic, highly reproducible, and cost-effective. Therefore, they have been widely used to assess genetic diversity among populations of a given taxon (*Gomes et al., 2020*; *Rohini et al., 2020*; *Senkoro et al., 2020*; *Xue et al., 2021*; *Eken et al., 2022*). The present study aimed to understand the morpho-genetic diversity, structure, and differentiation of *A. senegalensis* populations from Western (Benin) and Southern (Mozambique) Africa and the role of climate and genetic factors in shaping phenotypic variability. Specifically, we have assessed, (i) the genetic diversity, population structure, and differentiation; (ii) the morphological diversity, and structure; (iii) the overlapping between genetical and morphological clustering of individuals; and (iv) the relative importance of climate in the morphological variation.

## MATERIAL AND METHODS

### Study area

The study was carried out in Niassa Special Reserve (NSR), Mozambique (Southern Africa), and in the Sudanian zone, Benin (Western Africa), two locations where the species is best known and used. NSR is located in Northern Mozambique approximately between latitudes 12°8′40′N and 12°22′40′N; and longitudes 37°21′00′E and 37°45′00′E (Fig. 1). It covers approximately 42,000 km$^2$ and has been described as the largest protected area of Mozambique and the third largest in Africa (*Ryan et al., 2016*; *Mbanze et al., 2019*). Seventy-two percent of the total area of NSR is covered by dry Zambezian Miombo woodlands that are dominated by *Brachystegia spiciformis* Benth, *Brachystegia boehmii* Taub, and *Julbernardia globiflora* Benth. (*White, 1983*). The climate is tropical sub-humid, with a dry and relatively hot period between May and October. The annual rainfall is on average 900 mm per year increasing from the East (800 mm) to the West (1,200 mm). Temperature ranges between 20 and 30 °C (*Allan et al., 2017*). About 60,000 people are living inside the reserve and are concentrated around the two main villages of Mecula (Moz_MEC) in the East and Mavago (Moz_MAV) in the West, and along the main

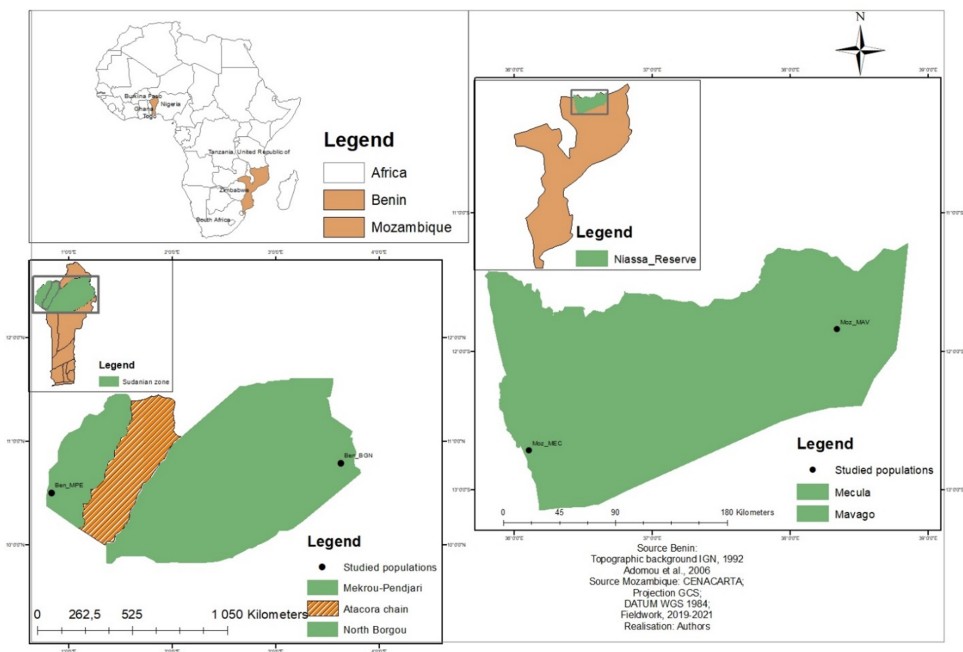

**Figure 1 Location of the studied populations in Benin and Mozambique.** Each data point indicates a sub-population. The brown colour indicates areas where no sub-population was sampled.

road (*NCP, 2017*; *SRN, 2008*; *Mbanze et al., 2021*). Slash-and-burn agriculture is the main livelihood activity of the population (*Cunliffe et al., 2009*).

The Sudanian zone is located in Northern Benin between latitudes 9°45′N and 12°25′N and longitudes 0°45′E and 3°55′E (Fig. 1) and is characterized by a tropical dry climate with two seasons (rainy and dry). The mean annual rainfall in this zone is often below 1,000 mm and the temperature is on average 27.5 °C (*Gnanglè et al., 2011*). The vegetation is composed of dry forests, woodlands, savannahs, and riparian forests. Common tree species in the area include *Isoberlinia* spp., *Combretum* spp., *Acacia* spp., *Hyparrhenia* spp., *Loudetia* spp., and *Andropogon* spp. (*Gnanglè et al., 2012*). North Borgou (Ben_BGN) and Mekrou pendjari (Ben_MPE) are the two main phytogeographical districts of the Sudanian zone of Benin. People living in the Sudanian zone of Benin are mainly farmers.

## Sampling and data collection

A total of 154 individuals of *Annona senegalensis* from the two geographical regions (Fig. 1; Table 1) were analyzed for genetic diversity and population structure. Due to the unavailability of trees bearing mature fruits in some populations, morphological data focused on a total of 147 individuals. In each region, two sub-populations were selected and within each, and leaves and fruits were collected along a linear transect of 30 km, with a minimum distance of 100 m to 10 km to avoid sampling siblings. Twenty-seven to sixty individuals were sampled within each sub-population and used for genetic and morphological analysis (Table 1). Samples from all individuals were brought to the laboratory for morphological analysis. For the genetic analysis, fresh leaves were kept

**Table 1  Study areas, populations, geographical coordinates and sample number.**

| Regions | Populations | Longitude | Latitude | Altitude | # samples |
|---|---|---|---|---|---|
| Western Africa (Benin) | Ben_BGN | 3.55885 | 10.93528 | 308 | 30 |
|  | Ben_MPE | 0.85198 | 10.51104 | 167 | 60 |
| Southern Africa (Mozambique) | Moz_MEC | 37.10937 | −12.07651 | 281 | 37 |
|  | Moz_MAV | 36.01857 | −12.08318 | 1087 | 27 |

Notes.
Ben_BGN, North Borgou population from Benin; Ben_MPE, Mekrou pendjari population from Benin; Moz_MEC, Mecula population from Mozambique; Moz_MAV, Mavago population from Mozambique.

in silica gel while in the field and stored at −80 °C once in the laboratory until DNA extraction.

## DNA extraction and nSSR amplification

Genomic DNA was isolated with the InnuSPEED Plant DNA Kit (Innuscreen GmbH, Analytik, Jena, Germany) according to the procedures described by the manufacturer, using 50 mg of ground leaves. The yield and purity of each sample was determined by spectrophotometry at 230, 260, and 280 nm (Nanodrop 2000, Thermo Fisher Scientific, Waltham, MA, USA), and complemented by agarose gel electrophoresis (*Gomes et al., 2020*). Based on the consistency of the polymorphic amplifications, ten microsatellite markers were used for genetic diversity analysis: LMCH4, LMCH6, LMCH11 (*Escribano, Viruel & Hormaza, 2004*), LMCH29, LMCH43, LMCH48, LMCH78, LMCH79, LMCH119, and LMCH122 (*Escribano, Viruel & Hormaza, 2008*).

PCR reactions were performed under the following conditions: 94 °C for 1 min; 94 °C for 30 s, 55 °C for 30 s (45 °C for LMCH29), 72 °C for 1 min (35 cycles); and 72 °C for 5 min. Each reaction was performed in a final volume of 15 μL containing 100 ng of genomic DNA, 0.4 μM each primer (Table 2), 1.25U MyTaq DNA polymerase and 1X MyTaq Reaction Buffer (Meridian Bioscience, Cincinnati, OH, USA). Forward primers were labeled with a fluorescent dye at the 5′-end. PCR products were separated by capillary electrophoresis on a CEQ™ 8000 capillary DNA analysis system (Beckman Coulter, Fullerton, CA, USA) and allele sizes were determined with GeneMapper 3.2 (Applied Biosystems, Waltham, MA, USA).

## Data on morphological traits

Data were collected in Mozambique from January to April 2021 and in Benin from June to September 2021, after fruiting. Six morphological descriptors were measured on plants, namely total height, bole height, crown height, trunk diameter at breast height, crown diameter, and crown shape. The bole height is the height from the ground to the first big branch and the crown height is the difference between total height (m) and bole height (m). The crown shape was derived from the ratio of crown height over crown diameter. To determine the crown diameter (m), four radii were measured from the projection of the crown on the ground (*Glèlè Kakaï et al., 2011*; *Hounkpèvi et al., 2016*). At least 40 leaves and 40 ripened fruits were collected per individual. Seven morphological fruit descriptors were measured: fruit length (mm), fruit width (mm), fruit dry weight (g), number of seeds per

**Table 2 Locus name, primer sequences, GenBank Accession number, and expected size of the amplified fragments from the polymorphic nSSR markers used in this study.**

| Name | Primer sequences (5′ − 3′) | Accession no. | Repeat | Size (bp) | Reference study |
|---|---|---|---|---|---|
| LMCH4 | F: ATTAGAACAAGGACGAGAAT R: CCTGTGTCTTTCATGGAC | AY685391 | $(GA)_{14}$ | 112–128 | *Escribano, Viruel & Hormaza (2004)* |
| LMCH6 | F: GGCATCCTATATTCAGGTTT R: TTAAACATTTTGGACAGACC | AY685393 | $(CT)_{14}$ | 220–254 | *Escribano, Viruel & Hormaza (2004)* |
| LMCH11 | F: TACCTCTCGCTTCTCTTCCT R: GATGATTAGACACAAGTGGATG | AY685398 | $(CT)_{10}$ | 173–176 | *Escribano, Viruel & Hormaza (2004)* |
| LMCH29 | F: GTACCATCTTTTAGGAAATC R: TGCAATCTATGTTAGTCAC | DQ923748 | $(GA)_{9}$ | 185–195 | *Escribano, Viruel & Hormaza (2008)* |
| LMCH43 | F: CTAGTTCCAAGACGTGAGAGAT R: ATAGGAATAAGGGACTGTTGAG | EF424144 | $(GA)_{9}$ | 210–216 | *Escribano, Viruel & Hormaza (2008)* |
| LMCH48 | F: TTAGAGTGAAAAGCGGCAAG R: TCAAGCTACAGAAAGTCTACCG | EF424148 | $(GA)_{12}$ | 141–154 | *Escribano, Viruel & Hormaza (2008)* |
| LMCH78 | F: ATTTGATTGATTGATTTCCTA R: CTTTTGCTTTCTTTCACATC | EF424169 | $(GA)_{9}$ | 159–161 | *Escribano, Viruel & Hormaza (2008)* |
| LMCH79 | F: GAAGCAAGTAGACACGTAGTA R: AGGGTTGGTATTTCTTTATAGT | EF424170 | $(CT)_{12}$ | 206–210 | *Escribano, Viruel & Hormaza (2008)* |
| LMCH119 | F: CAGAAAATTAGCAGAGGACTCA R: GTGGGTTGGGTTTTTAGGTC | EF424198 | $(GA)_{12}$ | 191–212 | *Escribano, Viruel & Hormaza (2008)* |
| LMCH122 | F: AGCAAAGATAAAGAGAAGATAA R: ATCCAAGCCTATTAACAACT | EF424200 | $(GA)_{9}$ | 177–210 | *Escribano, Viruel & Hormaza (2008)* |

fruit, seeds weight (g), pulp dry mass (g), fruit shape and the ratio fruit length to fruit width (*Hounkpèvi et al., 2016*; *Lawin et al., 2021*). Six quantitative descriptors were measured on leaves, including leaf length (cm), leaf width (cm), limb length (cm), petiole length (cm), leaf dry weight (g), and the ratio of leaf length to petiole length (*Sun et al., 2020*; *Mollick et al., 2021*). Fruits and leaves were further oven-dried at 105 °C until constant weight for the determinations of fruit dry weight, pulp mass, seeds dry weight, and leaf dry weight. After measuring the fruit dry weight, each fruit was split manually and the seeds were separated from the pulp. The number of seeds per fruit was then counted, and the seeds weight and pulp mass were weighed. Weights were measured using a 0.01 g precision scale while a centimeter ruler and a digital caliper with a 0.01 mm level of precision were used for all other measurements (Table 3).

## Bioclimatic data

Using the GPS coordinates of each individual in QGIS 3.16.2 (*QGIS Development Team, 2021*), bioclimatic data was extracted from the CHELSA (Climatologies at High resolution for the Earth's Land Surface Areas) database, considering the last data available over 30 years (1979-2013).

## Genetic diversity, population structure, and differentiation

For each geographical area and sub-population, genetic diversity was assessed by calculating the total number of alleles (Ta), mean number of alleles per locus (Na), Shannon's information index (H), mean expected heterozygosity (He), mean observed heterozygosity (Ho), inbreeding coefficient (FIS), and Polymorphism Information Content (PIC) using

**Table 3  Morphological traits of plants, fruits, and leaves of *A. senegalensis*.**

| Organs | Morphological descriptors | Short name | Units | Equipment/material used |
|---|---|---|---|---|
| Plant | Total height | Tot.hei | m | Suunto Clinometer |
| | Bole height | Bol.hei | m | Suunto Clinometer |
| | Crown height | Crown.hei | m | NA |
| | Trunk diameter at breast | DBH | cm | Electronic caliper |
| | Crown diameter | Crown.diam | m | Suunto Clinometer |
| | Crown shape | Crown.shp | NA | NA |
| Fruit | Fruit length | Fruit.leng | mm | 0.01 mm resolution digital caliper |
| | Fruit width | Fruit.wid | mm | 0.01 mm resolution digital caliper |
| | Fruit dry weight | Fruit.wei | g | 0.01 g sensitive balance |
| | Number of seeds per fruit | Fruit.nseeds | NA | NA |
| | Seeds weight | Seeds.wei | g | 0.01 g sensitive balance |
| | Pulp dry mass | Pulp.mass | g | 0.01 g sensitive balance |
| | Fruit shape | Fruit.shp | NA | NA |
| Leaves | Leaf length | Leav.len | cm | Centimetre rule |
| | Leaf width | Leav.wid | cm | Centimetre rule |
| | Limb length | Limb.len | cm | Centimetre rule |
| | Petiole length | Petiol.len | cm | Centimetre rule |
| | Leaf dry weight | Leav.wei | g | 0.01 g sensitive balance |
| | Ratio leaf length to petiole length | Leav.len_Petiol.len | NA | NA |

**Notes.**
NA, not applicable.

GenAlEx 6.51 (*Peakall & Smouse, 2012*). The Bayesian program STRUCTURE v.2.3.4 (*Pritchard et al., 2000*) was used to test whether any discrete genetic structure existed among samples. The analysis was performed assuming 1 to 10 genetic clusters ($K$) with ten replications per $K$. Models were run assuming ancestral admixture and correlated allele frequencies using run lengths of 300,000 interactions for each $K$ after 50,000 burn-in steps. The optimum $K$ was determined using STRUCTURE HARVESTER (*Earl & VonHoldt, 2012*), which identifies the optimal $K$ based on both the posterior probability of the data for a given K and the $\Delta$K (*Evanno, Regnaut & Goudet, 2005*). The results of the replicates at the best-fit $K$ identified by STRUCTURE were then post-processed using CLUMPP 1.1.2 (*Jakobsson & Rosenberg, 2007*). A Principal Coordinates Analysis (PCoA) was also constructed in GenAlEx 6.51 (*Peakall & Smouse, 2012*) to detect the genetic relatedness among individuals based on Nei's genetic distance. Analysis of molecular variance (AMOVA) was performed to quantify the partitioning of genetic variance between the geographical regions, as well as between and within sub-populations that showed genetic differentiation in STRUCTURE and PCoA. Each AMOVA was run with 10,000 permutations at 0.95 significance levels in Arlequin 3.11 (*Exoffier, Laval & Schneider, 2005*). The relationships between population pairwise Nei's genetic distances and linear geographical distances (isolation by distance) were examined with a Mantel test (*Mantel, 1967*) implemented in Arlequin 3.11 (*Exoffier, Laval & Schneider, 2005*) using the same permutation and significance levels.

## Morphological diversity and structuring

The morphological traits of 40 leaves and 40 fruits were measured per individual, and the respective averages were used for statistical analysis. Individual data was recorded for the remaining traits, i.e., total height, bole height, crown height, trunk diameter at breast height, crown diameter, and crown shape. The mean, standard error, and coefficient of variation of each morphological trait were calculated by population and sub-population. The coefficient of variation ($cv$ %) was used to assess the variability of each morphological trait, considering a $cv < 25\%$ an indicator of weak variability (*Reza et al., 2017*). A student $t$-test was first used to evaluate differences between Northern and Southern populations. Similarly, an analysis of variance was used to compare traits among the four sub-populations. The assumptions of normality and homoscedasticity required to run these tests were checked previously, using the Shapiro–Wilks test and the Levene test, respectively. When the violation of the assumption of normality was severe ($p < 0.01$), the corresponding non-parametric test (Mann–Whitney or Kruskal-Wallis) was applied. When the Analysis of Variance (ANOVA) indicated a significant difference, a SNK-test was applied as a multiple comparison test in the package AGRICOLAE (*De Mendiburu & Agricolae, 2020*) to separate means.

To assess the relationship between the morphological descriptors and the bioclimatic variables, a redundancy analysis (RDA) within the VEGAN package was carried out on the least square mean values of the morphological descriptors and bioclimatic variables (Table S, supplementary data). The RDA was first carried out separately for the Western, and the Southern sub-populations. Another RDA analysis was implemented with the merged populations. These RDA analyses were intended to assess whether the relative importance of the relationships between bioclimatic variables and morphological variation was similar for the two regions. All analyses were implemented in R statistical software version 4.1.2 (*R Core Team, 2021*).

# RESULTS

## Genetic diversity, structure, and differentiation

A total of 156 alleles were found among the 154 *Annona senegalensis* samples. The number of alleles varied from 27 in the Southern (Moz_MAV sub-population) to 55 in the Western (Ben_BGN sub-population) region (Table 4). The total number of alleles was significantly higher in the sub-populations sampled in the Western region than in the Southern region ($F = 3.23$, $p = 0.023$). This pattern was also observed in the average number of alleles ($F = 2.05$, $p = 0.001$), the Shannon Diversity Index ($F = 1.04$, $p = 0.021$), and the observed ($F = 4.24$, $p = 0.019$) and expected heterozygosity ($F = 4.47$, $p = 0.024$) (Table 4). The percentage of polymorphic loci was overall very high and showed the same pattern ($F = 3.39$, $p = 0.025$) *i.e.*, higher in the Western than in the Southern population (Table 4). FIS showed negative values in all sampled populations (Table 4) suggesting a heterozygosity higher than expected under the Hardy-Weinberg assumption. FIS values were lower in the Western than in the Southern population ($F = 1.29$, $p = 0.012$; Table 4).

The Bayesian clustering program STRUCTURE found the highest LnP(D) and $\Delta$K values for $K = 2$ differentiating the samples collected in Benin from the ones collected

**Table 4  Genetic diversity values (mean ± standard error) of *A. senegalensis* in Western and Southern Africa.**

| Populations | Ta | Na | H | Ho | He | FIS | PIC |
|---|---|---|---|---|---|---|---|
| **Benin (western)** | | | | | | | |
| Ben_BGN | 55 | 5.50 ± 0.41 | 1.43 ± 0.11 | 0.74 ± 0.06 | 0.71 ± 0.03 | −0.06 ± 0.10 | 100.00 ± 0.00 |
| Bem_MPE | 41 | 4.10 ± 0.44 | 1.10 ± 0.19 | 0.74 ± 0.08 | 0.61 ± 0.05 | −0.22 ± 0.08 | 100.00 ± 0.29 |
| **Mozambique (southern)** | | | | | | | |
| Moz_MEC | 33 | 3.30 ± 0.39 | 0.87 ± 0.14 | 0.52 ± 0.13 | 0.49 ± 0.08 | −0.02 ± 0.15 | 90.00 ± 8.04 |
| Moz_MAV | 27 | 2.70 ± 0.33 | 0.69 ± 0.15 | 0.58 ± 0.14 | 0.41 ± 0.09 | −0.26 ± 0.13 | 90.00 ± 4.00 |
| All | 156 | 3.90 ± 0.30 | 1.02 ± 0.13 | 0.64 ± 0.05 | 0.56 ± 0.04 | −0.14 ± 0.09 | 95.00 ± 2.89 |

Notes.

Ben_BGN, North Borgou population from Benin; Ben_MPE, Mekrou pendjari population from Benin; Moz_MEC, Mecula population from Mozambique; Moz; Ta, total number of alleles; Na, average number of alleles; H, average Shannon's diversity index; Ho, average observed heterozygosity; He, average expected heterozygosity; FIS, inbreeding coefficient; PIC, % of polymorphic loci.

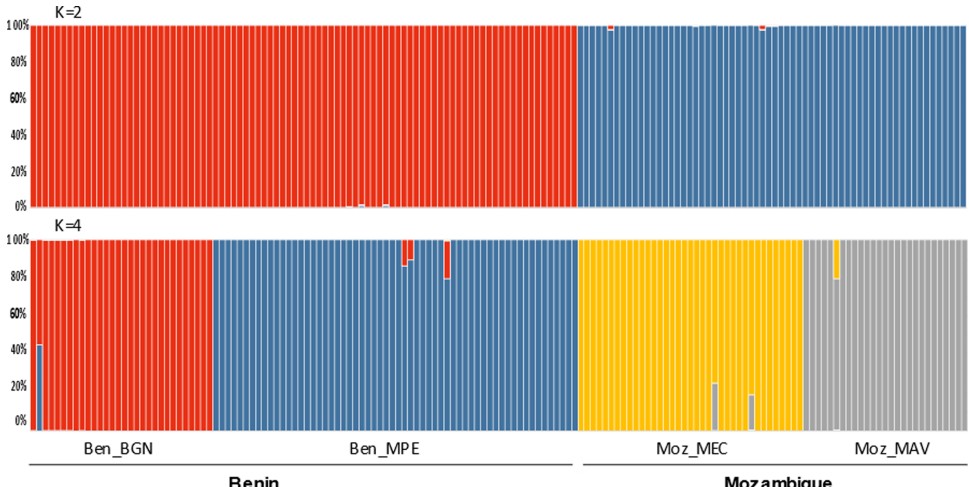

**Figure 2  Genetic structure of 154 *Annona senegalensis* samples collected in Benin and Mozambique, in four different populations: Borgou Nord (Ben_BGN), Mekrou Pendjari (Ben_MPE), Mecula (Ben_MEC) and Mavago (Ben_MAV).** Results are based on the best assignment results retrieved by STRUCTURE ($K = 2$ and $K = 4$). Each sample is represented by a thin vertical line divided into K-colored segments that represent the individual's estimated membership fractions in K clusters.

in Mozambique (Fig. 2). Nevertheless, STRUCTURE further revealed a secondary high LnP(D) and ΔK values at $K = 4$ differentiating the four sub-populations, Ben_BGN, Ben_MPE, Ben_MEC, and Ben_MAV into different genetic clusters (Fig. 2). Despite an overall high genetic integrity found in most samples, the results showed some signs of admixture between the genetic groups from Benin and Mozambique, although this admixture was negligible (Fig. 2). The same geographical pattern was retrieved by a principal coordinate analysis (PCoA) (Fig. 3). The first two coordinates of PCoA explained 35.9% of the total variation. Samples were spatially separated considering the two main geographic areas (Benin and Mozambique), but also by sub-populations following the $K = 4$ clustering result found in STRUCTURE (Fig. 2). The degree of spatial separation was lower for the two Mozambican sub-populations than for the ones from Benin (Fig. 3).

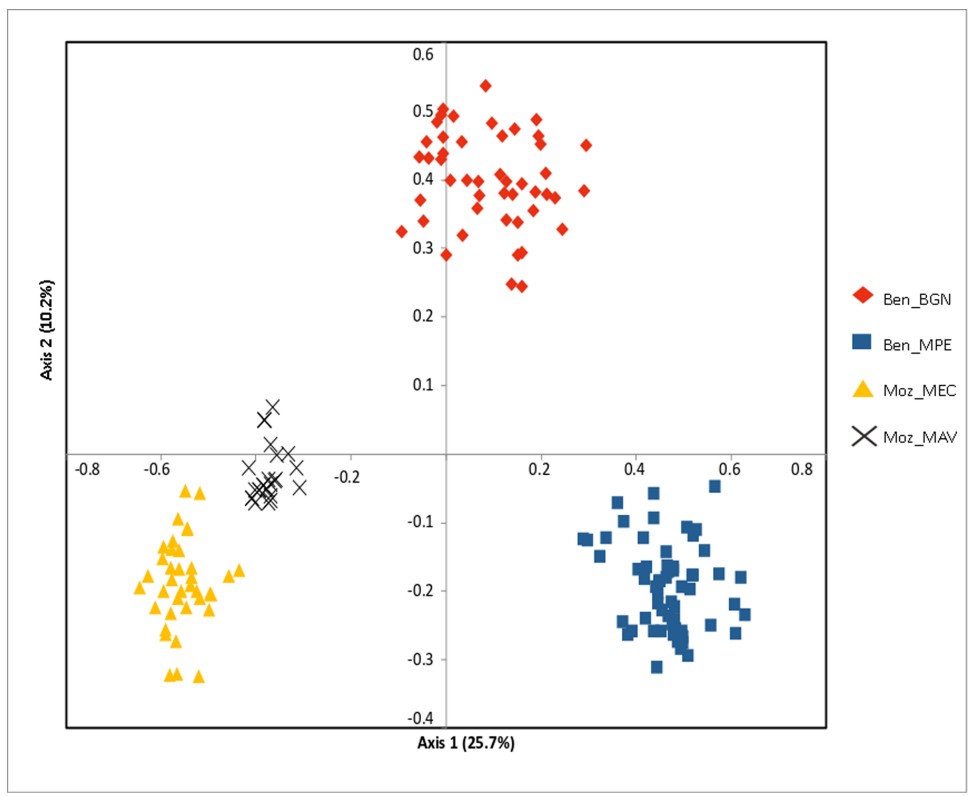

**Figure 3 Principal coordinate analysis (PCoA) of the studied *Annona senegalensis* populations.** The percentage of explained variance of each axis is given in parentheses. Population colors follow the $K = 4$ genetic groups identified by STRUCTURE.

AMOVA revealed that a high proportion of genetic variation was attributable to significant differences between the two regions (69.1%) supported by high levels of genetic differentiation (FST = 0.305, $p < 0.001$). In addition, 21.3% of variation occurred among populations while the remaining was found within sub-populations. In addition, the Mantel test confirmed the existence of a significant positive correlation between Nei's genetic distance and geographic distance for all pairwise sub-populations ($r = 0.212$, $p < 0.001$).

## Morphological diversity and structure

The morphological traits of *A. senegalensis* varied significantly between Northern and Southern populations. Plants from Southern populations were significantly larger (DBH: $15.89 \pm 2.10$ cm) and taller (Total height: $5.72 \pm 0.75$ m) than those from Northern populations (DBH: $5.89 \pm 0.62$ cm; Total height: $2.56 \pm 0.27$ m) (Table 5). Irrespective of the population, the coefficient of variation ($cv$) was high ($cv > 25\%$) for all traits. However, Northern populations had the highest $cv$ values irrespective of the traits, except for the crown shape. A significant difference was reported among sub-populations. However, the sub-population Ben_BGN (North Benin) had a similar trunk diameter at breast height (DBH) to that of Moz_MEC (South Mozambique). Within Southern populations,

**Table 5 Descriptive statistics of morphological traits of *A. senegalensis* plants individuals.**

| Parameters | Statistics | Populations | | Sub-populations | | | |
|---|---|---|---|---|---|---|---|
| | | Northern (Benin) | Southern (Mozambique) | Northern (Benin) | | Southern (Mozambique) | |
| | | | | Ben_BGN | Ben_MPE | Moz_MEC | Moz_MAV |
| DBH (cm) | m ± se | 5.89b ±0.62 | 15.89a ±2.10 | 10.15b ±1.85 | 3.76c ±0.48 | 12.17b ±4.22 | 20.03a ±3.85 |
| | cv (%) | 60.7 | 52.90 | 30.92 | 20.63 | 35.08 | 46.55 |
| | Min | 2.50 | 5.30 | 5.00 | 2.50 | 5.30 | 11.10 |
| | Max | 21.00 | 53.00 | 21.00 | 5.70 | 21.40 | 53.00 |
| Total height (m) | m ± se | 2.56b ±0.27 | 5.72a ±0.75 | 3.93c ±0.71 | 1.87d ±0.24 | 5.10b ±0.93 | 6.41a ±1.23 |
| | cv (%) | 45.17 | 31.04 | 15.49 | 33.96 | 23.31 | 32.27 |
| | Min | 1.10 | 2.60 | 2.30 | 1.10 | 2.60 | 3.90 |
| | Max | 4.80 | 10.40 | 4.80 | 4.50 | 7.50 | 10.40 |
| Bole height (m) | m ± se | 0.77b ±0.081 | 2.04a ±0.27 | 1.25b ±0.22 | 0.53c ±0.06 | 1.96a ±0.35 | 2.14a ±0.41 |
| | cv (%) | 53.74 | 39.78 | 27.38 | 29.63 | 28.67 | 48.06 |
| | Min | 0.27 | 0.70 | 0.45 | 0.27 | 0.70 | 0.80 |
| | Max | 1.90 | 5.00 | 1.90 | 1.00 | 3.00 | 5.00 |
| Crown height (m) | m ± se | 1.79b ±0.18 | 3.68a ±0.48 | 2.68b ±0.49 | 1.34c ±0.17 | 3.14b ±0.57 | 4.27a ±0.82 |
| | cv (%) | 47.56 | 45.21 | 22.54 | 41.5 | 36.18 | 45.69 |
| | Min | 0.70 | 0.90 | 1.10 | 0.70 | 0.90 | 1.30 |
| | Max | 3.80 | 9.20 | 3.65 | 3.80 | 5.00 | 9.20 |
| Crown diameter (m) | m ± se | 2.80a ±0.29 | 3.18a ±0.42 | 4.98a ±0.91 | 1.71b ±0.22 | 1.89b ±0.34 | 4.61a ±0.88 |
| | cv (%) | 65.55 | 65.39 | 32.13 | 27.84 | 19.62 | 49.07 |
| | Min | 0.88 | 0.96 | 2.30 | 0.88 | 0.96 | 1.48 |
| | Max | 7.99 | 10.80 | 7.99 | 3.13 | 2.56 | 10.80 |
| Crown size shape | m ± se | 0.73b ±0.07 | 1.44a ±0.19 | 0.58c ±0.10 | 0.80c ±0.10 | 1.72a ±0.31 | 1.14b ±0.21 |
| | cv (%) | 39.37 | 54.40 | 35.32 | 36.65 | 42.59 | 65.00 |
| | Min | 0.30 | 0.30 | 0.30 | 0.41 | 0.44 | 0.30 |
| | Max | 2.11 | 3.90 | 1.21 | 2.11 | 3.90 | 3.16 |

sub-populations Moz_MEC and Moz_MAV had similar values for bole height. The *cv* value decreased from both populations (31.04% to 65.55%) to sub-populations (15.49% to 65.00%), but was still relatively high. The DBH, bole height, crown height, crown diameter, and crown shape were more diverse in Moz_MAV, while the total height highly varied in Ben_MPE (Table 5).

The morphological parameters of fruits and leaves varied significantly among populations (Table 6). Considering the four sub-populations, results showed that Moz_MAV had the highest value for fruit length, fruit shape, fruit dry weight, number of seeds, seeds weight, pulp dry mass, limb length, leaf length, and leaf width. Ben_BGN had the highest fruit width, while the highest value for petiole length and leaf weight was recorded in Ben_MPE. Fruits from Mozambique were found to be bigger than those from Benin. Some traits, like the ratio of leaf length to petiole length, showed similar values between populations (Table 6). Furthermore, Ben_BGN and Moz_MEC presented similar values for fruit length and number of seeds per fruit. Both regions and their respective

sub-populations showed high *cv* regarding all traits, except fruit length, width, and fruit shape (Table 6). For fruit length, *cv* values varied from 15.53% to 15.96% in Western populations and from 14.53% to 18.76% in Southern populations. For fruit width, *cv* values varied from 11.68% to 13.43% in Western populations and from 12.46% to 14.58% in the Southern populations; and for fruit shape, *cv* values varied from 7.41% to 10.03% in Western populations and from 8.96% to 12.45% in Southern populations. The hierarchical clustering of the individuals based on their morphological traits resulted in four clusters (Fig. 4).

## Overlap between genetic and morphological clusters

The chi-square test was performed to test the association between morphological clusters and genetic clusters. Results (Pearson chi-square = 209.771, DF = 9, $p < 0.0001$; Likelihood ratio chi-square = 195.358, DF = 9, $p < 0.0001$) suggested a significant association between genetic and morphological variation, and hence, an effect of genetic factors on the morphological variation *i.e.*, the distribution of trees in genetic clusters is not independent of the morphological clusters. For instance, 86.67% of the individuals included in morphological cluster 1 corresponded to genetic cluster 1; and 70% of the individuals included in morphological cluster 3 correspond to the genetic cluster 3 (Table 7).

## Influence of bioclimatic variables on the morphological variation

The redundancy analysis showed that there was a significant correlation between morphological traits and bioclimatic variables. Furthermore, this relationship varied in diverse ways according to the two regions. In all cases, only the first two axes were significant ($p = 0.001$, $F = 12.489$) and explained the extent to which variation in morphological traits is related to bioclimatic variables. In Western populations, the model considered nine out of the 19 bioclimatic variables ($F = 7.7245$, $p = 0.001$, adjusted $R^2 = 0.404$). The first axis (RDA1) explained 80.77% of the total variance and was a combination of mean diurnal air temperature range (chelsa_b_1), temperature seasonality (chelsa_b_3), mean daily maximum air temperature in the warmest month (chelsa_b_4), annual range of air temperature (chelsa_b_6), mean daily air temperature of the wettest quarter (chelsa_b_7), mean daily air temperature in the warmest quarter (chelsa_b_9), and mean monthly precipitation in the coldest quarter (chelsa_b_18). The second axis (RDA2) explained 8.56% of the total variation and combined mean annual air temperature (chelsa_bio), and annual precipitation (chelsa_b_11).

In Southern populations, the model considered only five out of the 19 bioclimatic variables ($F = 5.3517$, $p = 0.001$, adjusted $R^2 = 0.279$. The first axis (RDA1) explained 67.17% of the total variance and was a combination of mean annual air temperature (chelsa_bio), mean diurnal air temperature range (chelsa_b_1), temperature seasonality (chelsa_b_3), and mean daily maximum air temperature in the warmest month (chelsa_b_4). The second axis (RDA2) explained 21.74% of the total variation and only considered the annual range of air temperature (chelsa_b_6).

When merging the two populations, the model considered 11 out of the 19 bioclimatic variables ($F = 12.489$, $p = 0.001$, adjusted $R^2 = 0.463$). The first axis (RDA1) explained

**Table 6 Descriptive statistics on morphological traits of fruits and leaves of *A. senegalensis*.**

| Parameters | Statistics | Populations | | Sub-populations | | | |
|---|---|---|---|---|---|---|---|
| | | Northern (Benin) | Southern (Mozambique) | Northern (Benin) | | Southern (Mozambique) | |
| | | | | Ben_BGN | Ben_MPE | Moz_MEC | Moz_MAV |
| Fruit length (mm) | m ± sd | 25.44b ±2.28 | 30.79a ±4.07 | 28.27b ±5.16 | 24.03c ±3.10 | 28.50b ±5.20 | 33.34a ±6.41 |
| | cv | 17.61 | 18.00 | 15.53 | 15.96 | 18.76 | 14.53 |
| | Min | 15.35 | 20.73 | 20.37 | 15.35 | 20.73 | 23.05 |
| | Max | 37.95 | 43.19 | 37.95 | 33.27 | 41.51 | 43.19 |
| Fruit width (mm) | m ± sd | 23.95b ±2.52 | 25.65a ±3.39 | 27.34a ±4.99 | 22.26c ±2.87 | 24.23a ±4.42 | 27.23a ±5.24 |
| | cv | 15.99 | 14.75 | 13.43 | 11.68 | 12.46 | 14.58 |
| | Min | 13.41 | 18.35 | 20.30 | 13.41 | 19.12 | 18.35 |
| | Max | 35.93 | 35.30 | 35.93 | 28.76 | 32.02 | 35.30 |
| Fruit shape | m ± sd | 1.06b ±0.11 | 1.21a ±0.16 | 1.03b ±0.18 | 1.08b ±0.13 | 1.20a ±0.21 | 1.23a ±0.23 |
| | cv | 9.47 | 10.88 | 7.41 | 10.03 | 12.45 | 8.96 |
| | Min | 0.90 | 1.01 | 0.93 | 0.90 | 1.01 | 1.02 |
| | Max | 1.49 | 1.62 | 1.23 | 1.49 | 1.62 | 1.50 |
| Fruit dry weight (g) | m ± sd | 2.42b ±0.25 | 5.95a ± 0.78 | 3.50c ±0.64 | 1.88d ±0.24 | 4.43b ± 0.80 | 7.63a ±1.46 |
| | cv | 59.52 | 38.06 | 50.70 | 44.62 | 34.08 | 22.32 |
| | Min | 0.63 | 1.91 | 1.54 | 0.63 | 1.91 | 3.99 |
| | Max | 10.93 | 10.10 | 10.93 | 4.83 | 7.98 | 10.10 |
| Number seeds (g) | m ± sd | 18.54b ±1.95 | 29.93a ±3.96 | 22.44b ±4.09 | 16.58c ±2.14 | 24.60b ±4.49 | 35.84a ±6.89 |
| | cv | 38.27 | 43.42 | 39.35 | 30.74 | 42.32 | 36.78 |
| | Min | 3.88 | 8.00 | 8.76 | 3.88 | 8.00 | 14.90 |
| | Max | 44.96 | 75.35 | 44.96 | 28.66 | 52.28 | 75.35 |
| Seeds weight (g) | m ± sd | 0.85b ±0.08 | 2.51a ±0.33 | 1.15c ±0.21 | 0.69d ±0.09 | 1.96b ±0.35 | 3.13a ±0.60 |
| | cv | 59.91 | 37.55 | 53.01 | 52.85 | 37.77 | 23.84 |
| | Min | 0.17 | 0.60 | 0.39 | 0.17 | 0.60 | 1.72 |
| | Max | 3.24 | 4.51 | 3.24 | 1.67 | 4.01 | 4.51 |
| Pulp dry mass (g) | m ± sd | 0.85b ±0.08 | 2.51a ±0.33 | 1.15c ±0.21 | 0.69d ±0.09 | 1.96b ±0.35 | 3.13a ±0.60 |
| | cv | 59.91 | 37.55 | 53.01 | 52.85 | 37.77 | 23.84 |
| | Min | 0.17 | 0.60 | 0.39 | 0.17 | 0.60 | 1.72 |
| | Max | 3.24 | 4.51 | 3.24 | 1.67 | 4.01 | 4.51 |
| Limb length (cm) | m ± sd | 9.83a ±1.03 | 9.04b ±1.19 | 9.50b ±1.73 | 10.00b ±1.29 | 7.26c ±1.32 | 11.01a ±2.11 |
| | cv | 17.82 | 25.73 | 1.72 | 19.26 | 18.21 | 12.89 |
| | Min | 1.19 | 5.23 | 6.78 | 1.19 | 5.23 | 8.36 |
| | Max | 13.94 | 14.34 | 12.29 | 13.94 | 10.04 | 14.34 |
| Leaf length (cm) | m ± sd | 7.10a ±0.74 | 6.66a ±0.88 | 6.54c ±1.19 | 7.38b ±0.95 | 5.00d ±0.91 | 8.50a ±1.63 |
| | cv | 17.63 | 34.66 | 13.33 | 17.91 | 34.53 | 14.24 |
| | Min | 4.23 | 2.54 | 4.65 | 4.23 | 2.54 | 6.17 |
| | Max | 11.37 | 11.77 | 8.20 | 11.37 | 9.27 | 11.77 |

**Table 6** (*continued*)

| Parameters | Statistics | Populations | | Sub-populations | | | |
|---|---|---|---|---|---|---|---|
| | | Northern (Benin) | Southern (Mozambique) | Northern (Benin) | | Southern (Mozambique) | |
| | | | | Ben_BGN | Ben_MPE | Moz_MEC | Moz_MAV |
| Leaf width (cm) | m ± sd | 3.82a ±0.40 | 3.58a ±0.47 | 3.57b ±0.65 | 3.94ab ±0.50 | 2.99c ±0.54 | 4.23a ±0.81 |
| | cv | 21.77 | 26.33 | 23.84 | 20.31 | 22.52 | 17.74 |
| | Min | 1.40 | 1.88 | 1.40 | 2.34 | 1.88 | 2.82 |
| | Max | 5.77 | 6.22 | 5.70 | 5.777 | 4.39 | 6.22 |
| Petiole length (cm) | m ± sd | 1.19a ±0.12 | 0.88b ±0.11 | 1.09a ±0.20 | 1.25ab ±0.16 | 0.76b ±0.14 | 1.02ab ±0.19 |
| | cv | 75.39 | 30.60 | 17.91 | 87.79 | 35.86 | 19.39 |
| | Min | 0.77 | 0.20 | 0.79 | 0.77 | 0.20 | 0.56 |
| | Max | 9.48 | 1.53 | 1.54 | 9.48 | 1.53 | 1.42 |
| Ratio leaf length to petiole length | m ± sd | 6.70b ±0.70 | 10.27a ±1.36 | 6.41b ±1.17 | 6.85b ±0.88 | 10.54a ±1.92 | 9.96a ±1.91 |
| | cv | 23.83 | 64.74 | 22.58 | 24.23 | 84.77 | 24.48 |
| | Min | 0.93 | 3.25 | 4.23 | 0.93 | 3.25 | 6.82 |
| | Max | 10.58 | 41.67 | 9.35 | 10.58 | 41.67 | 16.53 |
| Leaf weight (g) | m ± sd | 0.62a ±0.06 | 0.43b ±0.05 | 0.47c ±0.08 | 0.70a ±0.09 | 0.23d ±0.04 | 0.65b ±0.12 |
| | cv | 60.07 | 81.01 | 70.95 | 65.04 | 55.74 | 58.60 |
| | Min | 0.15 | 0.08 | 0.15 | 0.18 | 0.08 | 0.15 |
| | Max | 2.35 | 1.70 | 2.15 | 2.35 | 0.65 | 1.70 |

**Notes.**

Ben_BGN, North Borgou population from Benin; Ben_MPE, Mekrou pendjari population from Benin; Moz_MEC, Mecula population from Mozambique; Moz_MAV, Mavago population from Mozambique.

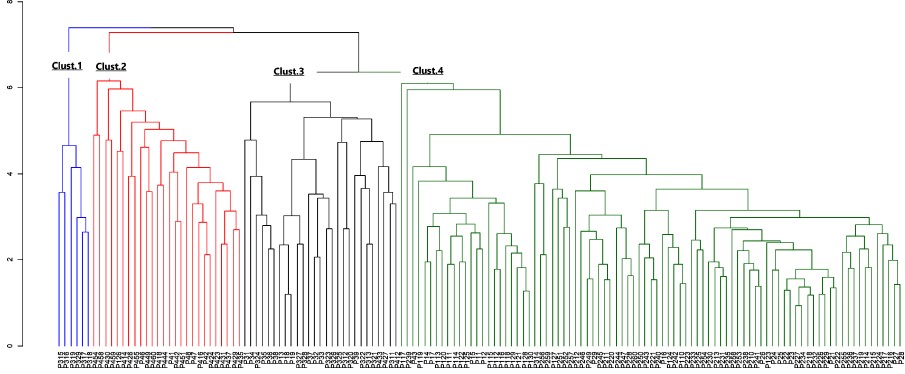

**Figure 4** **Dendrogram of morphological data.** The blue color indicates cluster 1, the red indicates cluster 2, the black indicates cluster 3 and the green, cluster 4.

**Table 7** Overlap between genetic and morphological clusters.

| Morpho | Gen_Cluster1 | Gen_Cluster2 | Gen_Cluster3 | Gen_Cluster4 |
|---|---|---|---|---|
| Morpho_Cluster1 | 86.67% | 3.33% | 10.00% | 0.00% |
| Morpho_Cluster2 | 98.33% | 1.67% | 0.00% | 0.00% |
| Morpho_Cluster3 | 3.33% | 6.67% | 70.00% | 20.00% |
| Morpho_Cluster4 | 3.70% | 85.19% | 11.11% | 0.00% |

83.44% of the total variance and was the combination of isothermality (chelsa_b_2); temperature seasonality (chelsa_b_3); mean daily minimum air temperature in the coldest month (chelsa_b_5); mean daily air temperature of the wettest quarter (chelsa_b_7); mean daily air temperature of the driest quarter (chelsa_b_8 ); annual precipitation (chelsa_b_11); amount of precipitation in the wettest month (chelsa_b_12); amount of precipitation in the driest month (chelsa_b_13); precipitation seasonality (chelsa_b_14); and mean monthly precipitation in the wettest quarter (chelsa_b_15). The second axis (RDA2) explained 6.19% of the variation and only considered the mean daily air temperature in the warmest quarter (chelsa_b_9). However, some of these bioclimatic variables such as mean annual air temperature, temperature seasonality, mean monthly precipitation in the coldest quarter, annual range of air temperature, and the mean monthly precipitation in the wettest quarter were not statistically significant in these different models (Table 8).

In the Western population, DBH, total height, bole height, crown height, crown diameter, crown shape, fruit length, fruit width, fruit shape, fruit dry weight, number of seeds per fruit, seeds weight, pulp dry mass, leaf width, petiole length, the ratio leaf length to petiole length, leaf dry weight and limb length were all loaded on RDA1, while only leaf length was loaded in RDA2 (Table 9). Based on the scores of morphological traits and bioclimatic variables on RDA axes (Tables 8, 9), leaf length was positively influenced by annual precipitation (chelsa_b_11). DBH, total height, bole height, crown height, crown diameter, fruit length, fruit width, fruit dry weight, number of seeds per fruit, seeds weight, and pulp dry mass were negatively influenced by mean diurnal air temperature range (chelsa_b_1), mean daily maximum air temperature in the warmest month (chelsa_b_4), annual range of air temperature (chelsa_b_6), and the mean daily air temperature of the wettest quarter (chelsa_b_7).

In Southern populations, all morphological parameters were loaded on RDA1 except bole height, crown diameter, crown shape, and the ratio leaf length to petiole length, which were loaded in RDA2. Crown diameter and the ratio leaf length to petiole length were negatively influenced by the mean annual air temperature (chelsa_bio), mean diurnal air temperature range (chelsa_b_1), temperature seasonality (chelsa_b_3), and the mean daily maximum air temperature in the warmest month (chelsa_b_4) (Tables 8, 9).

Considering the global set of sub-populations, morphological parameters were all loaded in RDA 1 except limb length, leaf width, the ratio leaf length to petiole length, and leaf weight. The DBH, total height, bole height, crown height, crown diameter, crown shape, fruit length, fruit width, fruit shape, fruit weight, number of seeds per fruit, seeds weight, pulp mass, and leaf length were positively influenced by the mean daily minimum air temperature in the coldest month (chelsa_b_5) and mean daily air temperature of the driest quarter (chelsa_b_8 ) (Tables 8, 9). The petiole length and the ratio leaf length to petiole length were negatively influenced by the isothermality (chelsa_b_2); temperature seasonality (chelsa_b_3); mean daily air temperature of the wettest quarter (chelsa_b_7); annual precipitation (chelsa_b_11) precipitation in the wettest month (chelsa_b_12); precipitation in the driest month (chelsa_b_13) precipitation seasonality (chelsa_b_14) mean monthly precipitation in the wettest quarter (chelsa_b_15) and the mean daily air temperature in the warmest quarter (chelsa_b_9), respectively (Tables 8, 9).

**Table 8  Significance of bioclimatic variables from permutation ANOVA test and scores on RDA axes.**

| Bioclimatic variable | Permutation Anova | | | | Scores on axis | |
|---|---|---|---|---|---|---|
| | Df | Variance | F | Pr (>F) | RDA1 | RDA2 |
| **Western populations** | | | | | | |
| chelsa_bio | 1 | 2.057 | 2.6707 | 0.062. ns | 0.310 | 0.378 |
| chelsa_b_1 | 1 | 5.625 | 7.3027 | 0.001*** | −0.238 | 0.162 |
| chelsa_b_3 | 1 | 2.633 | 3.4183 | 0.052 ns | −0.423 | −0.287 |
| chelsa_b_4 | 1 | 4.705 | 6.1084 | 0.004** | −0.555 | −0.090 |
| chelsa_b_6 | 1 | 5.108 | 6.6314 | 0.005** | −0.635 | 0.100 |
| chelsa_b_7 | 1 | 5.7 | 7.3998 | 0.003** | −0.685 | −0.185 |
| chelsa_b_9 | 1 | 3.782 | 4.9095 | 0.012* | 0.573 | 0.013 |
| chelsa__11 | 1 | 4.524 | 5.8731 | 0.009** | −0.209 | 0.325 |
| chelsa__18 | 1 | 1.965 | 2.5512 | 0.079. ns | 0.743 | 0.332 |
| Residual | 80 | 61.623 | | | | |
| **Southern populations** | | | | | | |
| chelsa_bio | 1 | 13.211 | 2.9011 | 0.043* | −0.776 | 0.303 |
| chelsa_b_1 | 1 | 13.244 | 2.9083 | 0.043* | −0.717 | 0.311 |
| chelsa_b_3 | 1 | 13.077 | 2.8716 | 0.043* | −0.947 | 0.106 |
| chelsa_b_4 | 1 | 12.875 | 2.8273 | 0.045* | −0.878 | −0.114 |
| chelsa_b_6 | 1 | 12.198 | 2.6787 | 0.052. ns | 0.053 | 0.535 |
| Residual | 51 | 232.246 | | | | |
| **All populations** | | | | | | |
| chelsa_b_2 | 1 | 8.434 | 8.2079 | 0.001*** | −0.835 | 0.244 |
| chelsa_b_3 | 1 | 6.8 | 6.6181 | 0.002** | −0.367 | −0.140 |
| chelsa_b_5 | 1 | 5.431 | 5.2851 | 0.007** | 0.505 | −0.256 |
| chelsa_b_7 | 1 | 8.876 | 8.6378 | 0.001*** | −0.352 | −0.045 |
| chelsa_b_8 | 1 | 7.183 | 6.9908 | 0.002** | 0.552 | −0.249 |
| chelsa_b_9 | 1 | 3.96 | 3.8541 | 0.024* | 0.068 | −0.199 |
| chelsa__11 | 1 | 3.762 | 3.661 | 0.019* | −0.732 | 0.282 |
| chelsa__12 | 1 | 5.285 | 5.1433 | 0.006** | −0.812 | 0.248 |
| chelsa__13 | 1 | 5.899 | 5.7414 | 0.006** | −0.515 | 0.218 |
| chelsa__14 | 1 | 8.476 | 8.2486 | 0.001*** | −0.781 | 0.250 |
| chelsa__15 | 1 | 2.205 | 2.1457 | 0.09. ns | −0.878 | 0.235 |
| Residual | 135 | 138.716 | | | | |

**Notes.**
ns, non-significant.
*$P$ value $< 0.05$.
**$P$ value $< 0.01$.
***$P$ value $< 0.001$.

## DISCUSSION

The genetic diversity in the studied populations was overall very high, with 156 alleles recorded among the 154 *Annona senegalensis* samples. This is much higher than the values reported by *Kwapata et al. (2007)* that found only a total of 23 alleles in 135 *A. senegalensis* samples collected in nine Malawi populations, and using a limited number of molecular markers. Heterozygosity values varied between 0.22 and 0.62 being attributed to changes

**Table 9** Scores of morphological traits on RDA axes.

| Morphological trait | Western population | | Southern population | | All populations merged | |
|---|---|---|---|---|---|---|
| | RDA1 (80.77%)*** | RDA2 (8.56%) | RDA1 (67.17%)*** | RDA2 (21.74%)* | RDA1 (83.44%) *** | RDA2 (6.19%). |
| DBH | −2.825 | −0.899 | 2.198 | −2.093 | 5.014 | −1.251 |
| Total height | −0.844 | −0.340 | 0.325 | 0.026 | 1.400 | −0.382 |
| Bole height | −0.265 | −0.156 | 0.053 | 0.137 | 0.502 | −0.147 |
| Crown height | −0.579 | −0.183 | 0.272 | −0.111 | 0.897 | −0.235 |
| Crown diameter | −1.404 | −0.558 | 0.689 | −0.818 | 0.837 | −0.265 |
| Crown size shape | 0.085 | 0.049 | −0.119 | 0.247 | 0.147 | −0.013 |
| Fruit length | −2.429 | −0.259 | 1.423 | 0.042 | 2.938 | −0.376 |
| Fruit width | −2.632 | −0.279 | 0.691 | −0.117 | 1.527 | −0.238 |
| Fruit shape | 0.014 | −0.001 | 0.017 | 0.003 | 0.051 | −0.010 |
| Fruit dry weight | −0.718 | −0.430 | 0.974 | −0.265 | 1.743 | −0.233 |
| Number of seeds per fruit | −3.664 | 1.514 | 4.573 | 1.404 | 6.331 | 1.653 |
| Seeds weight | −0.226 | −0.128 | 0.374 | −0.067 | 0.752 | −0.105 |
| Pulp dry mass | −0.226 | −0.128 | 0.374 | −0.067 | 0.752 | −0.105 |
| Limb length | 0.085 | 0.202 | 1.076 | −0.680 | 0.110 | 0.172 |
| Leaf length | 0.166 | 0.194 | 0.955 | −0.724 | 0.163 | 0.100 |
| Leaf width | 0.112 | 0.060 | 0.356 | −0.254 | 0.001 | 0.011 |
| Petiole length | 0.042 | 0.007 | 0.083 | −0.003 | −0.103 | 0.043 |
| Ratio leaf length to petiole length. | 0.126 | 0.000 | −0.426 | −1.506 | 0.964 | −1.188 |
| Leaf dry weight | 0.087 | 0.014 | 0.130 | −0.094 | −0.042 | 0.042 |

**Notes.**
*P value <0.05; **P value < 0.01; ***P value < 0.001; ns non-significant.

in population size and habitat heterogeneity. The values reported in this study were even higher than those observed in *Anona cherimola* Mill, another edible tree that has economic importance in many Mesoamerican countries, where wild and cultivated trees grow (*Escribano, Viruel & Hormaza, 2007*). Using 16 simple sequence repeat (SSR) loci in 279 *A. cherimoya* accessions from a worldwide *ex situ* field germplasm collection, *Escribano, Viruel & Hormaza (2007)* reported an average expected and observed heterozygosities of 0.53 and 0.44, respectively. An analysis of 20 *Annona* accessions belonging to four different species (*Annona reticulata* L., *Annona muricata* L., *Annona atemoya* Mabb., and *Annona squamosa* L.) collected from various locations and based on 11 RAPD and 12 SSRs markers, identified similar levels of heterozygosity. The high genetic diversity may be explained by protogynous dichogamy, a common breeding characteristic in Annonaceae, where female and male structures do not mature simultaneously (*González & Cuevas, 2011*). This mechanism prevents self-fertilization, encourages cross-pollination, and has clear implications for genetic diversity, both within and between species, but depends on the action of pollinators. For instance, in the Brazilian Cerrado, *Annona coriacea* Mart. has night anthesis producing a marked smell to attract several beetles that act as pollinators during the asynchronous female and male flowering periods (*Costa et al., 2017*). *Annona*
*crassiflora* Mart., another species from the Brazilian Cerrado also exhibits the same behavior, being the beetles responsible to promote cross-pollination, while visiting both female- and male-phase flowers (*Saravy, Marques & Schuchmann, 2022*). These breeding features would explain the high levels of genetic diversity found for *A. senegalensis* in this study.

Our results further showed a higher diversity and percentage of polymorphic loci in Western than in Southern populations. The center of origin of *A. senegalensis* could explain the higher level of diversity detected in Western populations. Although most *Annona* species are originated from South America and the Antilles, *A. senegalensis* is thought to have originated in Africa (*Pinto et al., 2005*). The species name is derived from Senegal (Western Africa) where the reference specimen was collected (*Lizana & Reginato, 1990*). Indeed, the number of specimens was predominant in Western populations. Small-size populations can lead to low heterozygosity values, which could imply inbreeding between trees as reported by several authors (*Angeloni, Ouborg & Leimu, 2011*; *Ellegren & Galtier, 2016*; *Rosenberger et al., 2021*). However, despite the lower FIS values observed in Western populations, they were negative in all sub-populations, suggesting a number of heterozygotes higher than expected according to the Hardy-Weinberg principle. This suggests the existence of gene flow between non-related individuals, supporting the cross-breeding pollination model described above.

Our results showed a high level of genetic differentiation in the studied populations. The Bayesian clustering program STRUCTURE presented the highest LnP(D) and $\Delta$K values for K = 2 differentiating the samples collected in the Western from those collected in the Southern region. STRUCTURE also revealed secondary high LnP(D) and $\Delta$K values at $K = 4$ differentiating the four sub-populations Ben_BGN, Ben_MPE (from the Western Region), Moz_MEC, and Moz_MAV (from the Southern region) into four different genetic clusters. These findings were supported by the PCoA which showed a clear spatial separation between Western and Southern populations and also between sub-populations within each geographical region. This genetic structure could be explained by the wide geographical distance that occurs between the two countries, which does not favour gene flow between them (*Yang et al., 2019*). This explains the high genetic differentiation found in AMOVA between the two regions (69.1%), than among (21.3%) or within (9.6%) sub-populations. Furthermore, the Mantel test confirmed the existence of a significant positive correlation between Nei's genetic distance and geographic distance for all pairwise sub-populations, suggesting that the geographical distribution contributed significantly to the observed genetic diversity. Still, some signs of genetic admixture were observed within sub-populations. Although floral heat production can attract pollinators over long distances, especially during the night (*Gottsberger, 1990*), beetles usually fly at close distances, which could explain the results found here. However, we should point that the degree of separation was lower between the two Southern sub-populations than the Western ones. This was probably due to a higher level of admixture in the former. Being from a protected area, and therefore from a relatively closed area with the availability of many dispersers, gene flow might be more facilitated in the studied Southern populations.

Likewise, high variation in morphological traits was observed between populations. Individuals from the two sub-populations from the Western region and the two from the

Southern region were grouped into four different clusters. The chi-square test performed on morphological and genetic data confirmed a significant association between the two, showing that the studied populations were morphologically and genetically connected, *i.e.*, the distribution of trees in genetic clusters was not independent of the morphological clusters. The strong association with genetic and morphological data might therefore reflect a high local adaptation of the species.

Morphological traits were also found to be highly influenced by the environment, mainly by temperature and precipitation indexes (Tables 8 and 9). In the Western region, the diameter at breast height (DBH), total height, bole height, crown height, crown diameter, fruit length, fruit width, fruit dry weight, number of seeds per fruit, seed weight, and pulp dry mass were negatively influenced by air temperature index, suggesting that increases in air temperature can lead to a reduction in those growth parameters. In the Southern region, similar trends were observed in temperature index, which negatively influenced other growth parameters such as the crown diameter and the ratio leaf length to petiole length. In the Western region, leaves were longer when the amount of annual precipitation increased. However, when combining both Western and Southern populations, petiole length was negatively influenced by some bioclimatic variables including the annual precipitation. This observation suggests that bioclimatic variables can have a contrasting effect on the morphological traits of the plants depending on the environmental conditions. These results suggest an important phenotypic plasticity in the surveyed *A. senegalensis* populations to different environments, in agreement with the findings of *Guerin, Wen & Lowe (2012)* that reported a morphological shift consistent with a response to contemporary climate change. However, the extent of the contribution of the environment should be further studied, since soil and other environmental variables such as topography were not included in the analysis (*Ouédraogo et al., 2019*). Therefore, studies considering data from additional environmental parameters are required to better estimate the effect of the genetic background vs. the environment on the variation of morphological traits of *A. senegalensis*. Yet the availability of high genetic diversity in the studied populations is a sign of biological adaptability that can enable *A. senegalensis* to respond in various ways to changes in the environment.

## CONCLUSIONS

The present study reported the morpho-genetic diversity in populations of *A. senegalensis* from Western (Benin) and Southern Africa (Mozambique). Strong differences were observed in the morphological traits scored in whole plants, fruits, and leaves between Western and Southern populations. Moreover, high genetic diversity was found in the studied populations. A significant association was found between morphological traits and genetic parameters. Precipitation and temperature extremes were found to be the most important climate factors, influencing *A. senegalensis* morphological traits. Our study provides crucial information for the sustainable management of this species.

## ACKNOWLEDGEMENTS

The authors are thankful to the Wildlife Conservation Society of Mozambique (WCS) for the cooperation and all the logistical support granted during our stay in Niassa Special Reserve. We acknowledge the determination and help of Dr Franziska Steinbruch and Mister Salimo Ndala during field activities.

### Funding

Janine Conforte Fifonssi Donhouedé was supported by the Regional Academic Exchange for Enhanced Skills in Fragile Ecosystems Management in Africa (REFORM AFRICA). The research work was supported by The Fund for Applied and Multisectoral Research (FIAM) from the Italian Cooperation and Eduardo Mondlane University and Fundação para a Ciência e a Tecnologia, through the contribution to the research unit UIDB/00239/2020 (CEF) and the Scientific Employment Stimulus - Individual Call (CEEC Individual) - 2021.01107. CEECIND/CP1689/CT0001 (Isabel Marques). The funders had no role in study design, data collection and analysis, decision to publish, or preparation of the manuscript.

### Grant Disclosures

The following grant information was disclosed by the authors:
The Regional Academic Exchange For Enhanced Skills in Fragile Ecosystems Management in Africa (REFORM AFRICA).
Applied and Multisectoral Research (FIAM) from the Italian Cooperation and Eduardo Mondlane University.
Fundação para a Ciência e a Tecnologia, through the contribution to the research unit: UIDB/00239/2020 (CEF).
The Scientific Employment Stimulus - Individual Call (CEEC Individual) - 2021.01107.: CEECIND/CP1689/CT0001.

### Competing Interests

Ana I. Ribeiro Barros is an Academic Editor for PeerJ.

### Author Contributions

- Janine Conforte Fifonssi Donhouedé conceived and designed the experiments, performed the experiments, analyzed the data, prepared figures and/or tables, authored or reviewed drafts of the article, and approved the final draft.
- Isabel Marques performed the experiments, analyzed the data, prepared figures and/or tables, authored or reviewed drafts of the article, and approved the final draft.
- Kolawolé Valère Salako performed the experiments, analyzed the data, prepared figures and/or tables, authored or reviewed drafts of the article, and approved the final draft.
- Achille Ephrem Assogbadjo conceived and designed the experiments, authored or reviewed drafts of the article, supervision, and approved the final draft.

- Natasha Ribeiro conceived and designed the experiments, authored or reviewed drafts of the article, supervision, and approved the final draft.
- Ana I.F. Ribeiro-Barros conceived and designed the experiments, performed the experiments, analyzed the data, prepared figures and/or tables, authored or reviewed drafts of the article, supervision, and approved the final draft.

### Data Deposition
The raw data used for morphological and genetic diversity study are available in the Supplementary Files.

### Supplemental Information
Supplemental information for this article can be found online at http://dx.doi.org/10.7717/peerj.15767#supplemental-information.

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
