# Peer review of "Genetic and morphological diversity in populations of Annona senegalensis Pers. occurring in Western (Benin) and Southern (Mozambique) Africa"

_PeerJ, doi:10.7717/peerj.15767_

## Round 0.1 · original submission · Minor Revisions

Please revise the article

Reviewer 1 ·

Basic reporting

The language of the manuscript conforms to professional standards. However, some typological and grammatical mistakes have been suggested in the manuscript that need to be corrected before further processing. The scientific names should be italics whenever mentioned in the manuscript. The same is missing in A. senegalensis (Line no.84, 139,251, 474). Similarly species names of the plants must be in italics format that need to be corrected throughout the manuscript where it marked in review panel and in reference list. Some repetitions have been made Line no. 259-261 and Line no. 417-419 that need to be corrected.

The literature cited are enough to justify the objective of the study. Some of the citations in the manuscript are missing in the reference list such as Reza et al., 2017 (Line no. 231), de Mendiburu, 2020 (Line no. 238). The year mentioned in the reference is mismatch with the year mentioned with the author in the list such as Hughes et al., 2016 in Introduction portion (Line no. 56) but in list it is 2008, Peakall and Smouse, 2012 (Line no. 212, 217) but in list 2006 and Pritchard et al., 2020 (Line no. 208) but in list 2000.Year of publication is missing in the list for Xue et al., 2021 (Line no. 101). Ellegren et al., 2016 (Line no. 415) should be Ellegren and Galtier 2016. Jakobsson et al., 2007 (Line no. 216) should be Jakobsson and Rosenberg 2007.
The tables and figures are clear and raw data has been shared. However, Table 1 title should also mention about the samples collected
Table No. 3 : Units of measurement in fourth column should be symbolized as per the standard format such as gram as “g” instead of “G”, centimeter as “cm” instead of “Cm” and so else others
Table 6 Descriptive statistics on fruit and leaf morphology among the individuals of A. senegalensis
Symbol of “cv” should be same not “Cv”
Leaves length (in table 6) and leaf length (written Line no. 303 and 307) ~ both should be unique
How can the Number seeds per fruit measured in gram unit as mentioned in the Table 6? Make necessary corrections.
Table 8 : ns non-significant *P value <0.05; *** P value <0.001
What about ** mentioned in the same column? Mention the note of the same (**) also.
Mention “NA” as not applicable at the end of the table.
Table 9 and figure 5 represents the same idea. Hence, one of these should be omitted before the final submission.
The authors of the article provided sufficient evidences to justify the hypothesis made by them.

Experimental design

The objectives of the study are clear and the theme of the article is related to the aims and scope of the journal.

The experiment methodology designed and utilized is appropriate. However, the authors need to mention that single individuals were studied and 40 fruits and leaf samples were taken and average values were used for analyses. The authors need to mention the study period (month and year) of the study. The authors need to clear that whether the duration when the vegetative traits (leaf traits) were measured r before flowering or after fruiting. All the individuals were studied at the same stage at the flowering, fruiting before and after fruiting because the leaf traits are also affected by the stage of the plant growth and development.

Validity of the findings

The research findings described here are novel and discussed thoroughly. The statistical methodology adopted for analyzing the results is robust and well described in the material. The hypothesis formed and conclusions derived from the study are sound. The authors well described and fulfills the requirements as stated in the objective of the study.

Additional comments

no comments

Annotated reviews are not available for download in order to protect the identity of reviewers who chose to remain anonymous.

·

Basic reporting

The text in the MS is quite clear and unambiguous. Authors have used professional English throughout the MS. However, minor modifications are needed, which are highlighted in the text itself.

Text and back reference patterns require revision. Kindly check the exact pattern and modify it accordingly for all the references. Authors can follow any previously published article in this journal.

MS is enriched with a good number of literature and sufficient field background/context is also provided. However, few more relevant and recently published literature can be cited in suitable MS places.

The authors have also shared the required structure, figures, tables, and raw data.

MS is also self-contained with relevant results to hypotheses.

Experimental design

MS also falls within the Aims and Scope of the journal.

Research questions are defined, relevant, and meaningful. It also identified the knowledge gap, particularly the molecular aspect of population diversity.

The investigation was performed to a high technical & ethical standard.

Methods are described with sufficient detail & information to replicate.

Validity of the findings

Impact and novelty not assessed.

Meaningful replication is encouraged where rationale & benefit to literature is clearly stated.

All underlying data have been provided; they are robust, statistically sound, & controlled.

Conclusions are well stated, linked to the original research question & limited to supporting results.

Additional comments

This is for the submitted manuscript (#81879) entitled “Genetic and morphological diversity in populations of Annona senegalensis Pers. occurring in Benin (western) and Mozambique (southern) Africa”
Annona is a well-known and important genus of fruit trees and has a quite good range of diversity. Particularly its wild-type species, Annona senegalensis, is a very popular fruit species having the enormous potential of addressing the nutritional, livelihood security, and other environmental issues in rural Africa.
The authors have well identified the research knowledge gap and attempted to fulfill it through molecular interventions/studies. The MS is well-written and included good content including tables, figures, and references wherever required. It follows the high technical & ethical standards as per the journal.
I recommend this MS to be published with minor modifications which are mentioned in the MS itself.

---

## Round 0.2 · Minor Revisions

Dear author

Your paper may be accepted if you can review the article considering the minor comments of reviewer.

Reviewer 1 ·

Basic reporting

The article meets the professional standards. However,
some typological and grammatical mistakes have been suggested in the manuscripts that need to be corrected before further processing. The scientific names should be Annona senegalensis Pers. in tables in figures which may be most imortant parts of citing the article. Overall suggested corrections have been made.

Experimental design

The article meets the aims and scopes of the journal and experimental material and methods are accurate as per my knowledge. The literature cited is enough to justify the objective of the study. No citation is missing however year missing in Hughes et al 2008 in the reference list that has been mentioned. However, the units should be mentioned in the article text that has been corrected. The crown shape to crown size and fruit shape to fruit size should be corrected in tables also speciallt Table 3 and 6.

Validity of the findings

The article meets the novelty parameter and results sound statistically more robust and justifiable. The conclusion is well stated the objectives and results of the article.

Additional comments

Some typographical mistakes mentioned in the text should be corrected before final publishing.

Annotated reviews are not available for download in order to protect the identity of reviewers who chose to remain anonymous.

·

Basic reporting

no comment

Experimental design

no comment

Validity of the findings

no comment

Additional comments

MS has been improved by the authors as per the comments suggested.
MS can be accepted for publication.

---

## Round 0.3 · accepted · Accept

Dear authors, your revised manuscript is accepted for publication.

Reviewer 1 ·

Basic reporting

The article meets the professional standards. Overall suggested corrections have been made. No comments

Experimental design

No comments

Validity of the findings

No comments

Additional comments

The article has been revised and no more comments for further revision is required as per my knowledge and understanding.